# Determination of the Minimum Sample Amount for Capillary Electrophoresis-Fourier Transform Mass Spectrometry (CE-FTMS)-Based Metabolomics of Colorectal Cancer Biopsies

**DOI:** 10.3390/biomedicines11061706

**Published:** 2023-06-13

**Authors:** Tetsuo Sugishita, Masanori Tokunaga, Kenjiro Kami, Kozue Terai, Hiroyuki Yamamoto, Hajime Shinohara, Yusuke Kinugasa

**Affiliations:** 1Department of Gastrointestinal Surgery, Tokyo Medical and Dental University, 1-5-45, Yushima, Bunkyo, Tokyo 113-8510, Japan; t.sugishita129@gmail.com (T.S.);; 2Human Metabolome Technologies, Inc., Tsuruoka 997-0052, Japan; kkami@humanmetabolome.com (K.K.);

**Keywords:** metabolomics, CE-FTMS, colorectal cancer

## Abstract

The minimum sample volume for capillary electrophoresis-Fourier transform mass spectrometry (CE-FTMS) useful for analyzing hydrophilic metabolites was investigated using samples obtained from colorectal cancer patients. One, two, five, and ten biopsies were collected from tumor and nontumor parts of the surgically removed specimens from each of the five patients who had undergone colorectal cancer surgery. Metabolomics was performed on the collected samples using CE-FTMS. To determine the minimum number of specimens based on data volume and biological interpretability, we compared the number of annotated metabolites in each sample with different numbers of biopsies and conducted principal component analysis (PCA), hierarchical cluster analysis (HCA), quantitative enrichment analysis (QEA), and random forest analysis (RFA). The number of metabolites detected in one biopsy was significantly lower than those in 2, 5, and 10 biopsies, whereas those detected among 2, 5, and 10 pieces were not significantly different. Moreover, a binary classification model developed by RFA based on 2-biopsy data perfectly distinguished tumor and nontumor samples with 5- and 10-biopsy data. Taken together, two biopsies would be sufficient for CE-FTMS-based metabolomics from a data content and biological interpretability viewpoint, which opens the gate of biopsy metabolomics for practical clinical applications.

## 1. Introduction

According to the World Health Organization, in 2020, colorectal cancer ranked third in incidence and second in deaths among malignant tumors [1]. Since colorectal cancer is commonly encountered in clinical practice, the analysis of cancer characteristics at the molecular level and the development of therapeutic agents based on these characteristics have proceeded at a faster pace than for other cancers. Fluorouracil (5-FU) was developed in 1957 to play a central role in the treatment of colorectal cancer [2]. In the 1990s, the efficacy of irinotecan and oxaliplatin was demonstrated [3,4,5,6], and after the 2000s, the usefulness of bevacizumab, cetuximab, panitumumab, ramucirumab, and other molecular targeting agents was demonstrated, expanding treatment options [7,8,9,10,11,12].

In recent years, the development of therapeutic agents for colorectal cancer has been dominated by the development of new molecularly targeted agents. The use of genomics, a type of omics, is essential for the development of these molecularly targeted drugs. Omics analysis includes genomics targeting DNA sequences, transcriptomics targeting RNAs, proteomics targeting proteins, and metabolomics targeting metabolites, among others. Genomics is currently the primary application field in clinical practice. However, limited information can be provided on the actual phenotypes since genomics is based on the analysis of upstream gene sequences in homeostasis. Conversely, metabolomics analyzes the most downstream metabolites in homeostasis and thereby allows a better understanding of signals directly associated with phenotypes. Thus, in the future, therapeutic agents targeting metabolites identified by metabolomics.

Several studies have shown metabolites and metabolic pathways characteristic of colorectal cancer using real tissues, and these metabolites are mostly measured by capillary electrophoresis (CE), liquid chromatography, and gas chromatography (GC) connected to mass spectrometry (MS) [13,14]. Among these, CE-MS is best suited for analyzing ionic metabolites, especially highly charged or phosphate compounds, the main components of energy metabolism in cancer. However, in these conventional methods, the required specimen amount is up to 20–40 mg, and the difficulty of obtaining a sufficient amount of specimen using nonsurgical means has greatly limited their clinical application. Therefore, capillary electrophoresis-Fourier transform mass spectrometry (CE-FTMS) has been developed and applied, showing approximately tenfold higher sensitivity than conventional CE connected to time-of-flight MS.

The development of CE-FTMS may enable the analysis with a smaller sample volume and actual clinical application in the future; however, the specific minimum amount of sample has not been clarified for CE-FTMS-based metabolome analysis. Therefore, this study aimed to evaluate the minimum amount of biopsied samples needed to ensure the quality of metabolomics by collecting colorectal tumor specimens and analyzing them with CE-FTMS.

## 2. Materials and Methods

### 2.1. Specimen Collection and Pretreatment

This study was approved by the ethical review committee of Tokyo Medical and Dental University Hospital (M2019-225). All biopsied samples were collected from five patients with colorectal cancer at Tokyo Medical and Dental University Hospital. Based on preoperative examination findings, we selected lesions with sufficient tumor volume that would not affect the diagnosis even if tumor tissue was collected. Tissue collection was initiated within 15 min after surgically removing the specimen. Using biopsy forceps for lower gastrointestinal endoscopy, 1, 2, 5, and 10 sites were taken from the tumor sites, and each was placed by batch. They were also similarly collected from the normal mucosa and placed in a batch. The collected tissues were frozen with liquid nitrogen in batches and stored in a freezer at ≤−80 °C until metabolome analysis.

### 2.2. Metabolite Extraction

Metabolite extraction and metabolome analysis were conducted at Human Metabolome Technologies, Inc. (HMT), Tsuruoka, Japan. Biopsied frozen tissue samples were weighed and placed in homogenization tubes along with zirconia beads (5 mm and 3 mm). Next, 50% of acetonitrile/Milli-Q water containing internal standards (H3304-1002, HMT, Tsuruoka, Yamagata, Japan) was added to the tubes, and samples were completely homogenized at 1500 rpm at 4 °C for 60 s using a bead shaker (Shake Master NEO, Bio-Medical Science, Tokyo, Japan). Then, the homogenate was centrifuged at 2300× *g* at 4 °C for 5 min. Subsequently, the upper aqueous layer was centrifugally filtered through a Millipore 5-kDa cutoff filter (UltrafreeMC-PLHCC, HMT) at 9100× *g* at 4 °C for 180 min to remove macromolecules. The filtrate was evaporated to dryness under a vacuum and reconstituted in Milli-Q water for metabolome analysis at HMT.

### 2.3. Metabolome Analysis

Metabolome analysis was conducted using HMT’s ω Scan package with CE-FTMS based on the previously described methods [15]. Briefly, CE-FTMS analysis was performed using an Agilent 7100 CE capillary electrophoresis system equipped with a Q Exactive Plus (Thermo Fisher Scientific Inc., Waltham, MA, USA), an Agilent 1260 isocratic HPLC pump, an Agilent G1603A CE-MS adapter kit, and an Agilent G1607A CE-ESI-MS sprayer kit (Agilent Technologies, Inc., Santa Clara, CA, USA). The systems were controlled by the Agilent MassHunter workstation software LC/MS data acquisition for 6200 series TOF/6500 series Q-TOF version B.08.00 (Agilent Technologies) and Xcalibur (Thermo Fisher Scientific) and connected by a fused silica capillary (50 μm i.d. × 80 cm total length) with commercial electrophoresis buffer (H3301-1001 and I3302-1023 for cation and anion analyses, respectively; HMT) as the electrolyte. The spectrometer was scanned from *m*/*z* 60 to 900 and from *m*/*z* 70 to 1050 in positive and negative modes, respectively [16]. Peaks with S/N > 3 were extracted using MasterHands 2.18.0.1, an automatic integration software (Keio University, Tsuruoka, Yamagata, Japan), to obtain peak information, including *m*/*z*, peak area, and migration time (MT) [16]. Signal peaks corresponding to isotopomers, adduct ions, and other product ions of known metabolites were excluded, and the remaining peaks were annotated based on their *m*/*z* values and MTs using HMT’s metabolite database, which was developed by running authentic chemical standards under the same analytical conditions. Areas of the annotated peaks were then normalized to internal standards and also by sample weights to obtain relative levels of each metabolite (Appendix A).

### 2.4. Statistical Analysis

Principal component analysis (PCA) [17] was performed using the HMT’s proprietary R program. Statistical significance was evaluated using Welch’s *t*-test, and detected metabolites were plotted on metabolite pathway maps using VANTED 2.1.0 software [18]. For subsequent data analyses, as a pre-processing of metabolome data, metabolites with missing values of ≥5 out of 10 samples were excluded for statistical analysis. By default, missing values were imputed by 1/5 of the minimum positive values of each detected metabolite, and metabolite levels were transformed to z values (mean-centered and divided by the standard deviations of each metabolite). Hierarchical clustering analysis was conducted using the MeV v4.9.0 software with Euclidean distance as the distance calculation method [19]. Quantitative metabolite set enrichment analysis (QMSEA) was performed using the MetaboAnalyst 5.0 software [20,21]. The Kyoto Encyclopedia of Genes and Genomes database was selected as the metabolite set library [22]. Random forest was performed with twofold cross-validation to make the binary classification model. The number of metabolites in each tree was optimized, and the number of decision trees for ensembles was set at 500. Metabolite selection was performed using recursive feature elimination and fivefold cross-validation. The importance of metabolites in the random forest model was measured by the mean decrease in accuracy. All computations regarding the random forest were performed using the caret package in R.

## 3. Results

### 3.1. Patient Characteristics

Patient characteristics are shown in Table 1. All five colorectal cancers were left-sided colorectal cancers, and four of these were well-differentiated adenocarcinomas.

### 3.2. Comparison of the Number of Metabolites Detected in Different Numbers of Biopsied Samples

Table 2 shows the number of metabolites detected in the different biopsy samples after mass correction. Figure 1 shows the tissue weight on the X-axis and the number of metabolites detected on the Y-axis. The average numbers of metabolites detected in 1, 2, 5, and 10 pieces of biopsied samples were 424 ± 34 (average ± SD), 458 ± 18, 455 ± 23, and 450 ± 22, respectively, and thus, in >2 pieces of biopsied samples, the number of detected metabolites was >450. As a result, the number of metabolites detected in 1 piece of the biopsied sample was significantly lower than that in 2, 5, and 10 pieces of the biopsied sample (vs. 2 pieces, *p* = 0.015; vs. 5 pieces, *p* = 0.027; vs. 10 pieces, *p* = 0.058). The number of metabolites detected among 2, 5, and 10 pieces was not significantly different.

The number of metabolites detected in the tumor and nontumor sites was not significantly different.

The following 13 substances were detected in ≥2 pieces of biopsied samples but not in one piece: 2,4-dichlorobenzoic acid, 2-amino-2-methyl-1,3-propanediol, 5-oxo-2-tetrahydrofurancarboxylic acid, betonicine, dATP, dCTP, digalacturonic acid, dTTP, isobutylamine, N-ethylglycine, oxamic acid, pyruvic acid, and sucrose 6’-phosphate.

### 3.3. PCA and Heat Maps

PCA showed that the tumor and nontumor sites were separated by the PC2 axis (Figure 2). Most samples from the same patient were plotted close to each other. However, A1-N1, A2-T1, and A3-N1 were separated by the PC1 axis but showed a similar trend to the separation of tumor and nontumor sites in the PC2 axis.

Figure 3A shows the heat map with clustering; the heat map suggests that metabolomic profiles in one piece of the biopsied sample tend to be different from those in other samples. Figure 3B shows the heat map created using only those metabolites detected in 2, 5, and 10 pieces of biopsies, showing a statistically significant difference between tumor and nontumor sites, when excluding data from one biopsy. Figure 3B visually shows metabolite differences between tumor and nontumor sites, with similar metabolite sets detected in each of the two sites.

### 3.4. Pathway Maps and QMSEA

Pathway maps were created for 2, 5, and 10 pieces of biopsied samples (Appendix A), and QMSEA was performed (Table 3). Among the pathways enriched, cysteine and methionine metabolism (*p* < 0.001), purine metabolism (*p* < 0.001), taurine and hypotaurine metabolism (*p* < 0.001), glycerophospholipid metabolism (*p* = 0.002), pyrimidine metabolism (*p* = 0.003), nitrogen metabolism (*p* = 0.004), tryptophan metabolism (*p* = 0.002), pyrimidine metabolism (*p* = 0.006), D-glutamine and D-glutamate metabolism (*p* = 0.010), fructose and mannose metabolism (*p* = 0.024), and propanoate metabolism (*p* = 0.034) showed significant differences between tumor and nontumor sites in all 2, 5, and 10 pieces of biopsied samples.

### 3.5. Random Forest

Finally, random forest analysis with twofold cross-validation was performed using 2-biopsy data to develop a binary classification model for distinguishing tumor and nontumor samples. As a result, 15 metabolites were selected as multi-metabolite markers based on their variable importance (Table 4). The top three metabolites, 5-hydroxyindoleacetic acid (5-Hydroxy-IAA), indoleacetaldehyde, and formylanthranilate, are all Trp metabolites. 5-Hydroxy-IAA was significantly lower in tumor tissues (*p* < 0.012), whereas formylanthranilate was rather higher (*p* < 0.037). Then, the classification model was applied to 5- and 10-biopsy data and predicted the tumor or nontumor status with 100% accuracy, suggesting the possibility that 2 biopsies could be sufficient for developing a classification model that can distinguish tumors and nontumors as accurately as when using 5 or 10 biopsies (Figure 4).

## 4. Discussion

In this study, metabolomic analysis of colorectal biopsies was performed using the newly developed CE-FTMS and examined for the minimum number of specimens required for analysis. The metabolic characteristics of colorectal cancer were also examined based on the results of the CE-FTMS analysis.

Results of the CE-FTMS analysis showed that the number of detected metabolites was equivalent if ≥2 pieces of biopsies were used. In addition, Figure 1 implies that we need roughly 5 mg or more to secure appropriate data in terms of the number of detected metabolites. The weight of one piece of biopsy, however, varies significantly (from 1.0 to 10.6 mg), and thus, there is a risk in using just one piece of biopsy sample for CE-FTMS-based metabolomics and biological interpretation. Not only the number of metabolites detected but also the metabolomic profiles also resembled each other among the data obtained by ≥2 biopsies. Indeed, in the heat map, the detected metabolite profiles in 2, 5, and 10 biopsies were similar. QMSEA, using the data from >2 biopsies, identified 10 common pathways enriched in tumor and nontumor comparisons. Previous studies have also shown that most of these 10 pathways are altered in colorectal cancer metabolism. Furthermore, the classification model developed based on 2-biopsy data perfectly predicted tumor or nontumor status when applied to the 5- and 10-biopsy data, suggesting that a crucial metabolite set for distinguishing two groups can be captured with 2-biopsy data. Thus, CE-FTMS can detect the same biological features as conventional analysis methods with a smaller sample amount, such as biopsy specimens. Therefore, the minimum number of biopsies required for CE-FTMS analysis was considered to be two pieces (average 8.2 ± 4.6 mg in the mass). Since previous studies using conventional methods required sample volumes of 50–100 mg, CE-FTMS, which can perform accurate analysis with an average sample volume of 8.2 mg, is considered very useful clinically [23,24].

Pathway map results are particularly important for the clinical application of metabolomic analysis results. In recent years, cancer metabolic pathways have been attracting attention in the fields of tumor markers and new drug development; however, many aspects of metabolic pathways in colorectal cancer are still unclear. Among the pathways that showed significant differences in this study, pathways particularly relevant to cancer metabolism will be discussed.

Random forest analysis generated a tumor versus nontumor classification model comprising 15 metabolites; however, interestingly, tryptophan metabolites occupied the top three in the list, which echoes the results obtained in QMSEA. Indeed, the top three metabolites, 5-hydroxyindoleacetic acid (5-Hydroxy-IAA), indoleacetaldehyde, and formylanthranilate, represent three major pathways in tryptophan metabolism: serotonin, indole, and kynurenine (Figure 5). In general, in cancer metabolism, indoleamine-2,3-dioxygenase (IDO)1, IDO2, and tryptophan-2,3-dioxygenase (TDO2) are activated in the first step of tryptophan degradation [25,26]. This phenomenon results in the accumulation of kynurenine, which suppresses T-cell differentiation and function and promotes immune tumor escape. This study showed that serotonin and indole pathways were enhanced in nontumor sites of the colon, whereas the kynurenine pathway was predominantly enhanced in tumor sites, suggesting the promotion of immune escape in the tumor regions.

In nitrogen metabolism (Appendix A), glutamine has reportedly been metabolized more than other nonessential amino acids in cancer cells [27]. In the present study, glutamine metabolism was enhanced in tumor parts, suggesting increased glutamate production. MYC and KRAS (G12D mutation) are thought to be involved in this glutamine metabolism. In colorectal cancer, regardless of the presence or absence of KRAS mutations, glutamine is absorbed into the cell to produce fatty acids, proteins, and nucleic acids essential for cell survival and growth [28]. To facilitate glutamine entering the cell and activating the TCA cycle, glutaminase must be activated to change glutamine to glutamate, and previous studies have shown that this reaction is enhanced in colon cancer [29]. In this study, this mechanism may have resulted in decreased glutamine and increased glutamate levels at the tumor site.

In purine and pyrimidine metabolism (Appendix A), these metabolic pathways may reflect the status of nucleic acid synthesis. In purine metabolism, both AMP and GMP were increased in tumor sites. In general, in adenosine metabolism, increased conversion of ATP to ADP and ADP to AMP implies increased energy expenditure. In guanosine metabolism, increased GMP also indicates a similar event. In adenosine metabolism, the AMP is increased at the tumor site, and in guanosine metabolism, the GMP is increased at the tumor site. This phenomenon may be due to the following two reasons: first, the synthesis of nucleotides at the tumor site may have increased energy consumption and enhanced conversion from ATP and ADP; second, the purine salvage pathway may have been enhanced at the tumor site, resulting in increased AMP and GMP production from adenine and guanine [30]. In pyrimidine metabolism, although no significant difference was observed in UMP between the tumor and nontumor sites, UDP and UTP were significantly enhanced in the tumor area. Therefore, RNA synthesis is also enhanced by pyrimidine metabolism.

In cysteine and methionine metabolism (Appendix A), the results suggest that cystathionine, a peripheral substance in the methionine circuit, is significantly higher at the tumor site. Furthermore, cysteine, its peripheral substance, was significantly enhanced at the tumor site in its conversion to cystine. The majority of malignant cells are in an oxidative state due to cellular metabolism changes caused by oncogenes. Oxidative stress at the tumor site may have enhanced the conversion from cysteine to cystine. The mean value of cysteine/cystine in this study was 0.02 in the tumor and 0.10 in the nontumor sites. A lower cysteine/cystine ratio indicates greater exposure to oxidative stress [31], and this feature is more likely observed in the tumor than in nontumor sites.

Overall, the fact that two biopsies are sufficient is clinically useful. For example, it is practically impossible to obtain a 20–40 mg specimen, which is required for TOFMS-based metabolome analysis during pretreatment endoscopy; however, two biopsies can be easily performed. The ability to analyze such a small amount of specimen eliminates the need to resect the tumor site for analysis, enabling clinical applications with less invasive and less expensive procedures. A future challenge is to make the analysis more convenient and immediate. If the time required for metabolome analysis is further reduced, making a quick and detailed diagnosis simply by analyzing metabolites in biopsy specimens from the tumor site is possible in the future. Furthermore, it would be clinically significant to make decisions in selecting future colorectal cancer drugs targeting metabolites with a small biopsy specimen collected endoscopically.

Several limitations should be considered in this study. First, because this is a pilot study, the number of patients is small. In particular, a larger number of patients are needed to examine metabolic pathways. Second, the study was limited to patients with colorectal cancer. Since the histological type, genotype, and grade of cancer differ depending on the primary site, further studies are needed for other types of cancer. Third, the specimens in this study were not taken directly from patients but from surgically resected colon or rectum tissues. Therefore, there may be some differences in the metabolites detected when compared to biopsy samples directly collected from living subjects.

## 5. Conclusions

This study clarified that CE-FTMS-based metabolomic analysis is feasible with a minimum of 2 biopsies (8.2 ± 4.7 mg) to obtain data that are comparable when using 5 (16.3 ± 5.0 mg) or 10 (47.3 ± 21.0 mg) biopsies, which was supported by the number of identified metabolites and biological interpretability tested by QMSEA and random forest analysis. This paves the way for biopsy-based clinical metabolomics for tumor characterization and patient stratification in the future.

## Figures and Tables

**Figure 1 biomedicines-11-01706-f001:**
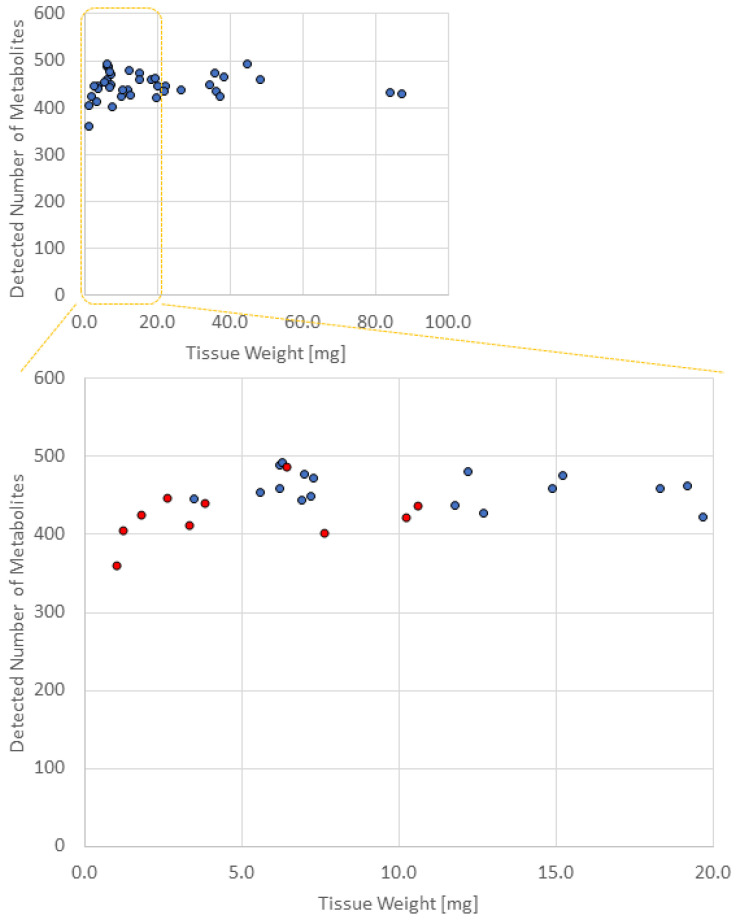
Number of metabolites detected in different weights of each sample. Red dots represent the samples with one biopsy count. Blue dots represent the samples with two, five, and ten biopsy counts. The number of metabolites detected in 1 piece of the biopsied sample was significantly lower than that in 2, 5, and 10 pieces of biopsied samples.

**Figure 2 biomedicines-11-01706-f002:**
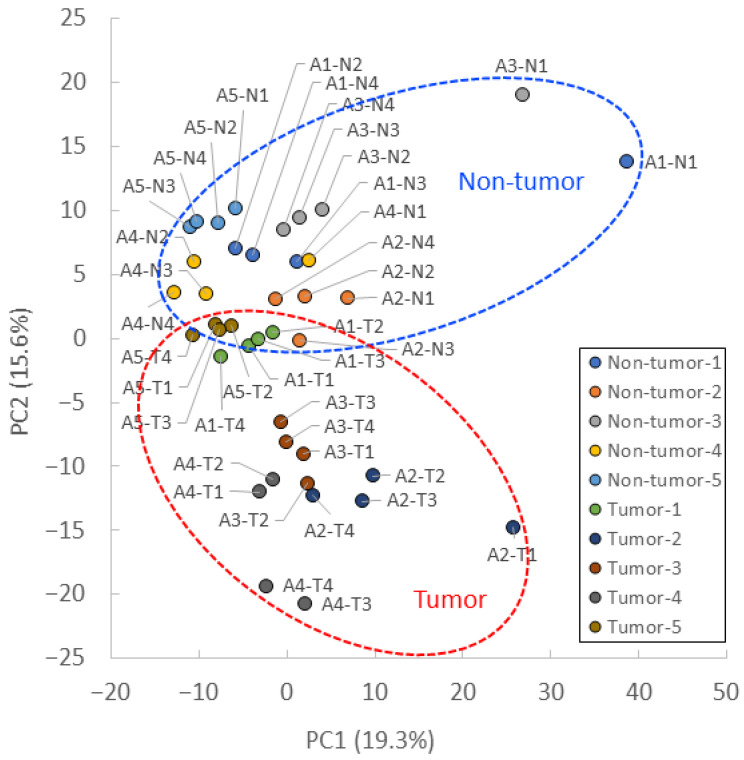
PCA plot of CE-FTMS metabolite profiles. PCA score plot for tumors and nontumors, with percentage variance for PC1 (19.3%) and PC2 (15.6%).

**Figure 3 biomedicines-11-01706-f003:**
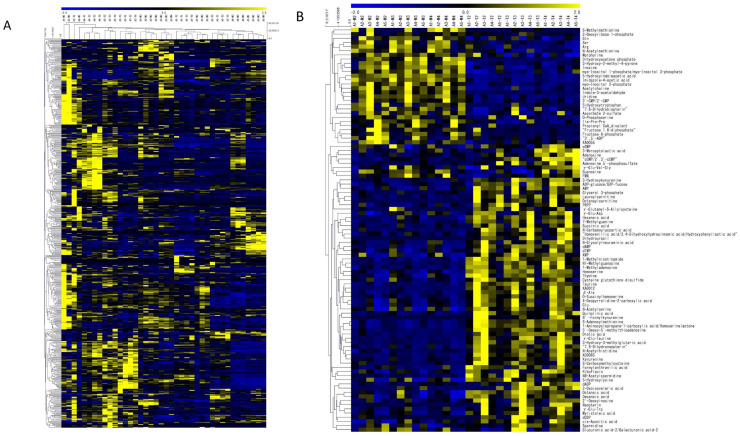
(**A**) Heat map with bi-clustered metabolomics data. (**B**) Heat map of metabolites with statistical significance between tumor and nontumor in any of 2, 5, or 10 pieces of biopsies.

**Figure 4 biomedicines-11-01706-f004:**
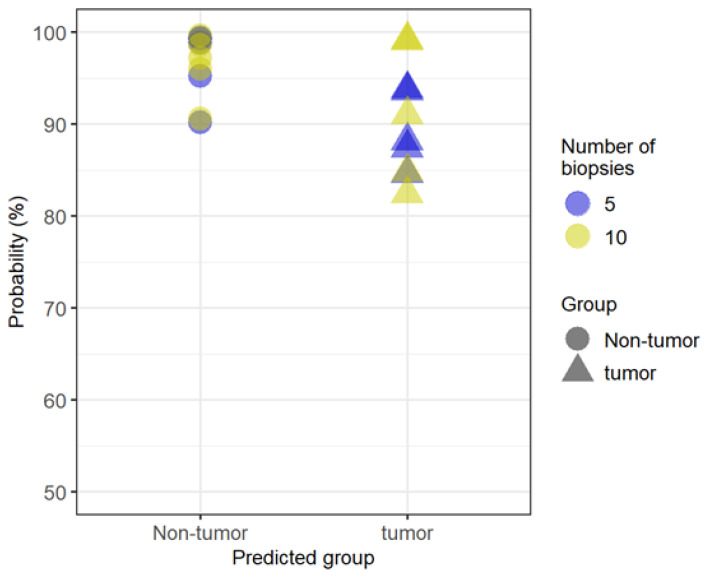
Probability scores of the classification model with 2-biopsy data applied to 5- and 10-biopsy data. The classification model predicted the tumor or nontumor status with 100% accuracy, suggesting the possibility that 2 biopsies could be sufficient for developing a classification model that can distinguish tumors and nontumors as accurately as when using 5 or 10 biopsies.

**Figure 5 biomedicines-11-01706-f005:**
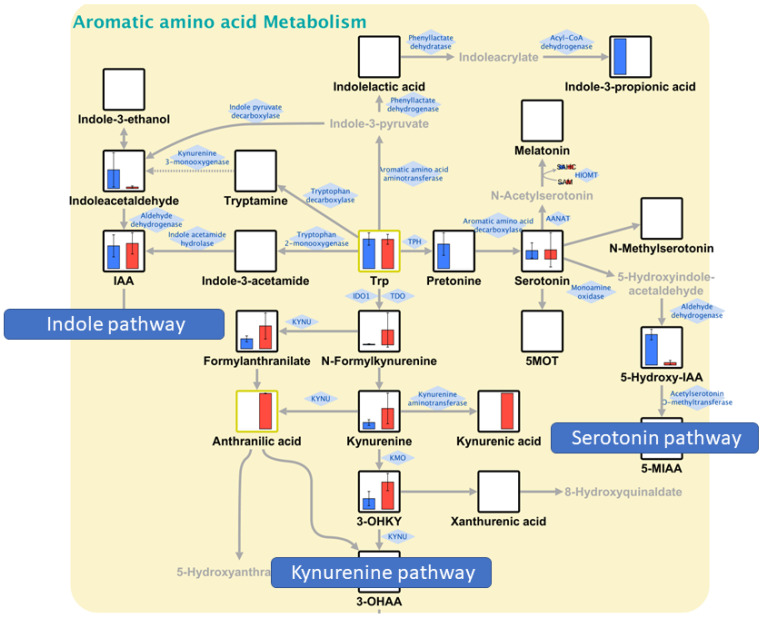
Tryptophan metabolism. Blue and red bars represent nontumor and tumor sites, respectively. Serotonin and indole pathways were relatively enhanced in nontumor sites of the colon, whereas the kynurenine pathway was predominantly enhanced in tumor sites.

**Table 1 biomedicines-11-01706-t001:** Tumor Characteristics.

Patient	A1	A2	A3	A4	A5
Age (years)	56	73	59	78	57
Gender	M	M	M	M	M
BMI (kg/m^2^)	25.5	25.5	30.7	21.3	23.3
Blood sugar (mg/dL)	108	104	131	114	99
AST (IU/L)	30	19	10	35	25
ALT (IU/L)	27	9	7	20	27
γ-GTP (IU/L)	178	21	19	134	59
Cr (mg/dL)	0.7	0.96	0.76	1.05	0.89
eGFR (mL/min/1.73 m^2^)	90.7	59.4	81.7	52.7	69.4
Tumor location	rectum	rectum	sigmoid colon	rectum	rectum
Tumor size (mm)	41 × 42	20 × 20	47 × 35	29 × 28	40 × 35
Histological type	tub2	tub1 > tub2	tub1 > pap	tub1 > tub2	tub1 > tub2
Depth of tumor invasion	T2	T1	T4	T4	T3
Lymphatic invasion	ly1	ly0	ly0	ly0	ly0
Venous invasion	v1	v1	v1a	v1a	v1b
Lymph node metastasis	N0	N0	N0	N0	N0

Abbreviations: BMI, body mass index; AST, aspartate aminotransferase; ALT, alanine aminotransferase; γ-GTP, γ-glutamyl transpeptidase; Cr, creatinine; eGFR, estimated glomerular filtration rate; tub1, well-differentiated tubular adenocarcinoma; tub2, moderately differentiated tubular adenocarcinoma; pap, papillary adenocarcinoma.

**Table 2 biomedicines-11-01706-t002:** Number of metabolites detected in different numbers of biopsied samples.

Sample Name	Number of Biopsies	Amount (mg)	Group Name	Detected Number of Metabolites
A1-N1	1	1.2	Nontumor-1	405
A1-N2	2	7.2		449
A1-N3	5	12.7		428
A1-N4	10	26.6		437
A2-N1	1	3.8	Nontumor-2	441
A2-N2	2	7.3		472
A2-N3	5	12.2		480
A2-N4	10	34.2		449
A3-N1	1	1.0	Nontumor-3	360
A3-N2	2	5.6		454
A3-N3	5	14.9		459
A3-N4	10	36.3		436
A4-N1	1	2.6	Nontumor-4	447
A4-N2	2	7.0		477
A4-N3	5	19.2		463
A4-N4	10	38.3		467
A5-N1	1	3.3	Nontumor-5	412
A5-N2	2	6.9		444
A5-N3	5	21.7		434
A5-N4	10	37.4		424
A1-T1	1	10.2	Tumor-1	423
A1-T2	2	6.2		459
A1-T3	5	19.7		422
A1-T4	10	87.2		431
A2-T1	1	1.8	Tumor-2	425
A2-T2	2	3.5		446
A2-T3	5	6.2		489
A2-T4	10	35.7		473
A3-T1	1	7.6	Tumor-3	403
A3-T2	2	11.8		437
A3-T3	5	22.4		445
A3-T4	10	48.4		459
A4-T1	1	6.4	Tumor-4	488
A4-T2	2	6.3		493
A4-T3	5	15.2		475
A4-T4	10	44.9		494
A5-T1	1	10.6	Tumor-5	438
A5-T2	2	20.1		446
A5-T3	5	18.3		459
A5-T4	10	84.0		433

**Table 3 biomedicines-11-01706-t003:** Comparison of statistically significant metabolic pathways enriched by QMSEA between tumor and nontumor in 2, 5, and 10 pieces of biopsies.

(1) N2 vs. T2 (Two Pieces of Biopsied Samples)	Raw *p*	(2) N3 vs. T3 (Five Pieces of Biopsied Samples)	Raw *p*	(3) N4 vs. T4 (Ten Pieces of Biopsied Samples)	Raw *p*	Statistical Significance
Taurine and hypotaurine metabolism	1.47 × 10^−3^	Nitrogen metabolism	1.24 × 10^−3^	Cysteine and methionine metabolism	9.28 × 10^−5^	(1)–(3)
Tryptophan metabolism	2.57 × 10^−3^	Tryptophan metabolism	2.43 × 10^−3^	Purine metabolism	9.48 × 10^−4^	(1)–(3)
Cysteine and methionine metabolism	4.04 × 10^−3^	Pyrimidine metabolism	3.83 × 10^−3^	Taurine and hypotaurine metabolism	1.19 × 10^−3^	(1)–(3)
Histidine metabolism	5.44 × 10^−3^	Glyoxylate and dicarboxylate metabolism	7.01 × 10^−3^	Glycerophospholipid metabolism	1.82 × 10^−3^	(1)–(3)
Folate biosynthesis	7.52 × 10^−3^	Purine metabolism	7.16 × 10^−3^	Pyrimidine metabolism	2.97 × 10^−3^	(1)–(3)
Nitrogen metabolism	8.20 × 10^−3^	D-glutamine and D-glutamate metabolism	8.87 × 10^−3^	Nitrogen metabolism	3.71 × 10^−3^	(1)–(3)
Pyrimidine metabolism	1.29 × 10^−2^	Fructose and mannose metabolism	1.06 × 10^−2^	Tryptophan metabolism	3.79 × 10^−3^	(1)–(3)
Riboflavin metabolism	1.30 × 10^−2^	Propanoate metabolism	1.49 × 10^−2^	D-glutamine and D-glutamate metabolism	9.56 × 10^−3^	(1)–(3)
D-glutamine and D-glutamate metabolism	1.46 × 10^−2^	Cysteine and methionine metabolism	2.04 × 10^−2^	Sulfur metabolism	1.37 × 10^−2^	(3) only
Purine metabolism	1.95 × 10^−2^	Inositol phosphate metabolism	2.17 × 10^−2^	Glycerolipid metabolism	1.42 × 10^−2^	(3) only
Propanoate metabolism	1.98 × 10^−2^	Beta-alanine metabolism	2.27 × 10^−2^	Glyoxylate and dicarboxylate metabolism	1.48 × 10^−2^	(2), (3) only
Glycerophospholipid metabolism	2.15 × 10^−2^	Amino sugar and nucleotide sugar metabolism	2.35 × 10^−2^	Histidine metabolism	2.22 × 10^−2^	(1), (3) only
Fructose and mannose metabolism	2.21 × 10^−2^	Taurine and hypotaurine metabolism	2.46 × 10^−2^	Fructose and mannose metabolism	2.42 × 10^−2^	(1)–(3)
Butanoate metabolism	2.25 × 10^−2^	Starch and sucrose metabolism	3.36 × 10^−2^	Folate biosynthesis	2.72 × 10^−2^	(1), (3) only
Pantothenate and CoA biosynthesis	2.91 × 10^−2^	Alanine, aspartate, and glutamate metabolism	3.50 × 10^−2^	Primary bile acid biosynthesis	2.93 × 10^−2^	(3) only
Arginine and proline metabolism	2.94 × 10^−2^	Neomycin, kanamycin, and gentamicin biosynthesis	3.70 × 10^−2^	Propanoate metabolism	3.38 × 10^−2^	(1)–(3)
Porphyrin and chlorophyll metabolism	3.36 × 10^−2^	Fatty acid biosynthesis	3.70 × 10^−2^	Sphingolipid metabolism	3.45 × 10^−2^	(3) only
Citrate cycle (TCA cycle)	4.07 × 10^−2^	Glycerophospholipid metabolism	4.12 × 10^−2^	Alanine, aspartate, and glutamate metabolism	3.69 × 10^−2^	(2), (3) only
One carbon pool by folate	4.18 × 10^−2^	Galactose metabolism	4.91 × 10^−2^	Nicotinate and nicotinamide metabolism	4.50 × 10^−2^	(3) only
				Arginine and proline metabolism	4.79 × 10^−2^	(1), (3) only
				Amino sugar and nucleotide sugar metabolism	4.80 × 10^−2^	(2), (3) only

**Table 4 biomedicines-11-01706-t004:** List of 15 metabolites selected by random forest analysis as multi-metabolite markers based on their variable importance.

Metabolite	Importance
5-Hydroxyindoleacetic acid	100
Indoleacetaldehyde	69.60
Formylanthranilate	67.28
XA0012	66.93
1-Methylnicotinamide	62.38
Taurine	62.02
Octanoylcarnitine	58.29
γ-Glu-Taurine	55.30
β-Ala	49.10
5′-Deoxy-5′-methylthioadenosine	42.98
1-Aminocyclopropane-1-carboxylic acid homoserine lactone	42.30
O-Succinylhomoserine	42.13
Myo-inositol 2-phosphate	36.75
Imidazole-4-acetic acid	34.39
Uridine	0

## Data Availability

The data presented in this study are available upon request from the corresponding author due to ethical concerns.

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
