# Peer review of "Determination of the Minimum Sample Amount for Capillary Electrophoresis-Fourier Transform Mass Spectrometry (CE-FTMS)-Based Metabolomics of Colorectal Cancer Biopsies"

_biomedicines, 2023, doi:10.3390/biomedicines11061706_

Round 1

Reviewer 1 Report

In the manuscript “Determination of the minimum sample amount for capillary electrophoresis-Fourier transform mass spectrometry (CE-FTMS)-based metabolomics of colorectal cancer biopsies”, the authors proposed to study the minimum amount of biopsied samples that ensure the quality of metabolomics by collecting colorectal tumor specimens and analyzing them with CE-FTMS. The study is timely and has interest. The paper is merely descriptive but there is no other way to perform these experiments. I have only very minor issues although the patients characterization is a critical matter. I have some specific comments that may be useful.

Specific comments:

1. Patients anthropometric characteristics must be well presented. Please include age, metabolic characteristics and other information regarding the selected patients.

2. Study limitations should be included. Please discuss the limitations of this study.

Only minor changes required

Author Response

Thank you for your valuable feedback.

1. Patients anthropometric characteristics must be well presented. Please include age, metabolic characteristics and other information regarding the selected patients.

→Added to table1.

2. Study limitations should be included. Please discuss the limitations of this study.

→The limitations are described from line 325. However, we have added the following note to enrich the content: “Third, the specimens in this study were not taken directly from patients, but from surgically resected colon or rectum tissues. Therefore, there may be some differences in the metabolites detected when compared to biopsy samples directly collected from living subjects.”

Reviewer 2 Report

In the submitted manuscript entitled “Determination of the minimum sample amount for capillary electrophoresis-Fourier transform mass spectrometry (CE- FTMS)-based metabolomics of colorectal cancer biopsies” Sugishita et al. investigated the amount of the minimum sample volume for capillary electrophoresis-Fourier transform mass spectrometry (CE-FTMS) useful for analyzing hydrophilic metabolites from samples obtained from colorectal cancer patients. The manuscript is well-written and fit the scope of the journal. The conducted experiments are appropriate. The results substantiate the conclusions. However, it should also be noted, that the number of patient is very small. Further studies are necessary to prove the statement and results.

Remarks

Page 2 line 50-51 Citations are necessary to this section.

It should be noted that there are some HPLC methods that are able to separate hydrophilic components and can hyphenate FTMS.

Page 3 line 110 It is very important to define what is the peak. Do you set any S/N ratio? What was your blank? Etc.

Please add more comments for Table 4. Is it enough to determine the level of 5-Hydroxyindoleacetic acid?

Author Response

Thank you for your valuable feedback.
#Page 2 line 50-51 Citations are necessary to this section.
→The lines 50-51 in the original document refer to the sentence beginning with “These metabolites are mostly measured…”, which is the description of citations 14 and 15; therefore, we combined the sentence with the immediately preceding one and added the citations 14 and 15 at the end.  

# It should be noted that there are some HPLC methods that are able to separate hydrophilic components and can hyphenate FTMS.  
 →We are aware that LC-MS using HILIC column, for example, is suitable for analyzing hydrophilic compounds, but CE-MS is even better at analyzing highly charged compounds or those reactive with metals such as phosphate compounds, and thus, we added “especially highly charged or phosphate compounds” in line 55.  

 # Page 3 line 110 It is very important to define what is the peak. Do you set any S/N ratio? What was your blank? Etc.  
 →We extracted peaks with S/N>3, so added a phrase, “with S/N>3”, in line 108. We didn’t set any blank sample; thus, a noise level was determined from the blank signals at the early minutes of each run.  

 # Please add more comments for Table 4. Is it enough to determine the level of 5-Hydroxyindoleacetic acid?  
 →We obtained only relatively quantified data (peak area of each metabolite with respect to that of internal standard in each sample), but added the following sentences from line 212: “The top 3 metabolites, 5-hydroxyindoleacetic acid (5-Hydroxy-IAA), indoleacetaldehyde, and formylanthranilate are all Trp metabolites. 5-Hydroxy-IAA was significantly lower in tumor tissues (p < 0.012), whereas formylanthranilate was rather higher (p < 0.037).”

Reviewer 3 Report

The authors present a paper that deals with the use of capillary  electrophoresis-Fourier transform mass spectrometry (CE-FTMS) to undertake metabolomics on colon cancer biopsies. I think the use of these new mass spectrometric techniques is an important topic in metabolomic studies and so I found the current manuscript very interesting.

From my perspective the paper is well written and close to a publishable standard

I have a few points for the authors to consider alongside those of ther other reviewers and editors

Although I realise that this paper is proof of concept I feel that the abstract could mention something about the significance of being able to get a metabolomic profile from 2 biopsy samples.

In any proof of concept study it is important for the readers to understand the next steps in development - from reading the paper I found that the message concerning a) the next stages of development and b) what this actually could mean for diagnosing patients (what are the benefits of metabolomic profiling vs. other conventionally used techniques) did not come through strongly enough.   Could the authors add slightly morer in the discussion/future work concerning this aspect?

I notice that the mass of biopsy samples is quite variable when looking at the table. This means that there is not always a clear distinction between 1 and 2 biopsy samples. Perhaps the authors should discuss this aspect in more detail and what it potentially means for clinical applications and metabolomic studies. I assume this is the main reason that 1 biopsy sample is not considered good enough although the total number of metabolites identified is not that different. Could the authors also comment on this aspect as if the numbers are not thnat different why is there this differentiation in the type of metabolites with 2 samples.

I feel that figure 3 could be formatted slightly better in sizing the respective heatmaps

I feel that the supplementary figures looking at the differences in pathways etc. are quite relevant to the paper and not sure why all of them are in supplementary information. It may be good to present one or more of these alongside the discussion

The authors don't really explain fully the process whereby metabolites are identified. I believe in metabolomic studies it is important to describe this process and also whether these were confirmed with standards etc. or if not how they were identified or annotated etc. I thought it was also the case that the raw data of all the identified metabolites should be made available via supplementary material

Author Response

Thank you for your valuable feedback.

# Although I realise that this paper is proof of concept I feel that the abstract could mention something about the significance of being able to get a metabolomic profile from 2 biopsy samples.
  →We added a phrase in line 25 of the abstract regarding the significance for clinical applications: “which opens the gate of biopsy metabolomics for practical clinical applications.” (The added phrase is minimal due to the 200 words limit)

# In any proof of concept study it is important for the readers to understand the next steps in development - from reading the paper I found that the message concerning a) the next stages of development and b) what this actually could mean for diagnosing patients (what are the benefits of metabolomic profiling vs. other conventionally used techniques) did not come through strongly enough.   Could the authors add slightly morer in the discussion/future work concerning this aspect?
 →The next steps and future perspectives are mentioned in lines 298-308 in the original manuscript, but we elaborated the paragraph by adding the following sentences in lines 317-320 for better clarity: The ability to analyze such a small amount of specimen eliminates the need to resect the tumor site for analysis, enabling clinical applications with less invasive and less expensive procedures. A future challenge is to make the analysis more convenient and immediate.

# I notice that the mass of biopsy samples is quite variable when looking at the table. This means that there is not always a clear distinction between 1 and 2 biopsy samples. Perhaps the authors should discuss this aspect in more detail and what it potentially means for clinical applications and metabolomic studies. I assume this is the main reason that 1 biopsy sample is not considered good enough although the total number of metabolites identified is not that different. Could the authors also comment on this aspect as if the numbers are not thnat different why is there this differentiation in the type of metabolites with 2 samples.
 →The reviewer’s comment is agreeable indeed and the weight of a single biopsy varies significantly, which is indeed a part of the reason why we concluded that two samples would be necessary. So, we added the following sentences from line 236: “In addition, Figure 1 implies that we need roughly about 5 mg or more to secure appropriate data in terms of the number of detected metabolites. The weight of one piece of biopsy, however, varies significantly (from 1.0 to 10.6 mg), and thus, there is a risk in using just one piece of biopsy sample for CE-FTMS-based metabolomics and biological interpretation.”

# I feel that figure 3 could be formatted slightly better in sizing the respective heatmaps  
→The page in Figure 3 is turned horizontally and enlarged.

# I feel that the supplementary figures looking at the differences in pathways etc. are quite relevant to the paper and not sure why all of them are in supplementary information. It may be good to present one or more of these alongside the discussion
→Since the tryptophan metabolic pathway is particularly important in our study, we decided to include it as Fig. 5. The others are listed as Supplementary Materials.

# The authors don't really explain fully the process whereby metabolites are identified. I believe in metabolomic studies it is important to describe this process and also whether these were confirmed with standards etc. or if not how they were identified or annotated etc. I thought it was also the case that the raw data of all the identified metabolites should be made available via supplementary material
 →The way we annotated metabolites is described in the end of “Metabolome analysis” section, but we elaborated the sentences as follows from line 110: “Signal peaks corresponding to isotopomers, adduct ions, and other product ions of known metabolites were excluded, and the remaining peaks were annotated based on their m/z values and MTs using HMT’s metabolite database, which was developed by running authentic chemical standards under the same analytical conditions.” In addition, the raw data to submit as Supplementary Table was prepared, which was mentioned at the end of the “Metabolome analysis” section in line 116 as Table S1.